# The Diversity of Methylation Patterns in Serous Borderline Ovarian Tumors and Serous Ovarian Carcinomas

**DOI:** 10.3390/cancers16203524

**Published:** 2024-10-18

**Authors:** Laura A. Szafron, Roksana Iwanicka-Nowicka, Piotr Sobiczewski, Marta Koblowska, Agnieszka Dansonka-Mieszkowska, Jolanta Kupryjanczyk, Lukasz M. Szafron

**Affiliations:** 1Maria Sklodowska-Curie National Research Institute of Oncology, 02-781 Warsaw, Poland; 2Laboratory of Systems Biology, Faculty of Biology, University of Warsaw, 02-106 Warsaw, Poland; r.iwanicka@uw.edu.pl (R.I.-N.); marta@ibb.waw.pl (M.K.); 3Laboratory for Microarray Analysis, Institute of Biochemistry and Biophysics, Polish Academy of Sciences, 02-106 Warsaw, Poland; 4Department of Gynecological Oncology, Maria Sklodowska-Curie National Research Institute of Oncology, 02-781 Warsaw, Poland; sobiczewskipiotr7@gmail.com; 5Cancer Molecular and Genetic Diagnostics Department, Maria Sklodowska-Curie National Research Institute of Oncology, 02-781 Warsaw, Poland; agnieszka.dansonka-mieszkowska@nio.gov.pl; 6Department of Cancer Pathomorphology, Maria Sklodowska-Curie National Research Institute of Oncology, 02-781 Warsaw, Poland; jolanta.kupryjanczyk@nio.gov.pl

**Keywords:** serous ovarian carcinoma, serous borderline ovarian tumor, DNA methylation, methylation microarrays, biomarkers

## Abstract

**Simple Summary:**

In tumorigenesis, aberrant DNA methylation may be an earlier and stronger modifier of gene expression than mutations. Herein, 128 serous ovarian tumors were analyzed, including borderline ovarian tumors (BOTS) with (BOT.V600E) and without (BOT) the *BRAF* V600E mutation, low-grade (lg), and high-grade (hg) ovarian cancers (OvCa). The methylome of the samples was profiled with Infinium MethylationEPIC microarrays. Global, genome-wide hypomethylation positively correlated with the increasing aggressiveness of tumors, being the strongest in hgOvCa. Remarkably, the ten most significant differentially methylated regions (DMRs) in the genome, discriminating BOT from lgOvCa, encompassed the MHC region on chromosome 6. We also identified hundreds of DMRs potentially useful as predictive biomarkers in BOTS and hgOvCa. DMRs with the best discriminative capabilities overlapped the following genes: *BAIAP3*, *IL34*, *WNT10A*, *NEU1*, *SLC44A4,* and *HMOX1*, *TCN2*, *PES1*, *RP1-56J10.8*, *ABR*, *NCAM1*, *RP11-629G13.1*, *AC006372.4*, *NPTXR* in BOTS and hgOvCa, respectively. By identifying potential biomarkers, this study might improve ovarian tumor outcome.

**Abstract:**

**Background**: Changes in DNA methylation patterns are a pivotal mechanism of carcinogenesis. In some tumors, aberrant methylation precedes genetic changes, while gene expression may be more frequently modified due to methylation alterations than by mutations. **Methods**: Herein, 128 serous ovarian tumors were analyzed, including borderline ovarian tumors (BOTS) with (BOT.V600E) and without (BOT) the *BRAF* V600E mutation, low-grade (lg), and high-grade (hg) ovarian cancers (OvCa). The methylome of the samples was profiled with Infinium MethylationEPIC microarrays. **Results**: The biggest number of differentially methylated (DM) CpGs and regions (DMRs) was found between lgOvCa and hgOvCa. By contrast, the BOT.V600E tumors had the lowest number of DM CpGs and DMRs compared to all other groups and, in relation to BOT, their genome was strongly downmethylated. Remarkably, the ten most significant DMRs, discriminating BOT from lgOvCa, encompassed the MHC region on chromosome 6. We also identified hundreds of DMRs, being of potential use as predictive biomarkers in BOTS and hgOvCa. DMRs with the best discriminative capabilities overlapped the following genes: *BAIAP3*, *IL34*, *WNT10A*, *NEU1*, *SLC44A4,* and *HMOX1*, *TCN2*, *PES1*, *RP1-56J10.8*, *ABR*, *NCAM1*, *RP11-629G13.1*, *AC006372.4*, *NPTXR* in BOTS and hgOvCa, respectively. **Conclusions**: The global genome-wide hypomethylation positively correlates with the increasing aggressiveness of ovarian tumors. We also assume that the immune system may play a pivotal role in the transition from BOTS to lgOvCa. Given that the BOT.V600E tumors had the lowest number of DM CpGs and DMRs compared to all other groups, when methylome is considered, such tumors might be placed in-between BOT and OvCa.

## 1. Introduction

Changes in DNA *methylation patterns are* a pivotal mechanism of carcinogenesis. In tumors, aberrant DNA methylation may be an earlier event than mutations. In some cancers, gene expression may even be more frequently modified due to methylation alterations than by mutations [1,2]. Borderline ovarian tumors (BOTS) exhibit intermediate aggressiveness between benign tumors and invasive carcinomas. They are a rare entity with relatively low malignant potential. In contrast to the majority of ovarian carcinomas, BOTS usually occur in women in reproductive age, are usually diagnosed at the low FIGO stage, and are characterized by better survival rates. Imaging methods (ultrasound, MRI) are useful to distinguish BOTS from OvCa preoperatively. However, the final diagnosis must be based on histopathological examination. Surgery with a complete resection is the cornerstone of BOTS treatment. Still, in young women considering procreation, a fertility-sparing surgical intervention is preferentially applied. Remarkably, chemotherapy is not recommended in BOTS [3,4]. Following the complete removal of the tumor, even 20% of BOTS may recur, usually as borderline tumors; however, in some patients, BOTS may recur as ovarian carcinomas [5,6,7,8]. Moreover, serous BOTS are closely related to serous low-grade carcinomas (lgOvCa), as they harbor similar genetic alterations [7,9]. By contrast, high-grade serous ovarian carcinomas (hgOvCa) are considered distinct ovarian neoplasms, molecularly unrelated to lgOvCa and BOTS [10]. Considering methylome changes, it was reported that serous hgOvCa form a separate cluster compared to BOTS and lgOvCa [11]. However, so far, methylation patterns of BOTS and lgOvCa of the serous type have been evaluated with low-resolution microarrays only. In addition, scientific data comparing ovarian tumors of diverse aggressiveness are still very scarce [11,12,13,14]. To fill out this gap, we aimed to obtain very detailed methylation profiles in such tumors. For this purpose, we performed the methylation analysis in non-consecutive primary serous ovarian tumors, obtained from previously untreated patients, using high throughput microarrays. This analysis was then validated by methylation-specific PCR combined with Sanger sequencing. Moreover, to investigate the biological role of the nominated biomarkers and assess their clinical usefulness, we carried out detailed DNA strand-specific and fold change-dependent ontology analyses, followed by the comprehensive statistical inference of all differentially methylated regions in the genome with multivariable regression models. Considering that the TP53 accumulation status has been previously shown to affect the clinical meaning of other molecular markers in our previous research on ovarian cancers [15,16], we decided to take this parameter into account in the present study. As to BOTS, we investigated these tumors in the context of the mutational status of the *BRAF* oncogene, which was demonstrated to be crucial for borderline ovarian tumors but not ovarian cancers [17,18]. Simultaneously, the presence of the *BRAF* V600E mutation turned out to be a negative clinical factor, associated with the earlier onset of BOTS in our previous research [19].

## 2. Materials and Methods

### 2.1. Patients and Clinicopathological Parameters

In the present study, a set of 128 non-consecutive, primary serous ovarian tumors of different aggressiveness was investigated. All the patients with these tumors were hospitalized at the Maria Sklodowska-Curie National Research Institute of Oncology, Warsaw, Poland in the years 1995–2015. Medical records of the patients were critically reviewed by at least two physicians. Our set of tumors included 25 BOTS (11 with and 14 without the *BRAF* V600E mutation, Appendix A) and 103 OvCa (7 lgOvCa and 96 hgOvCa, Appendix A). The specimens were selected to meet the following criteria: adequate staging procedure according to the recommendations by the International Federation of Gynecologists and Obstetricians (FIGO), tumor tissue from the first laparotomy available, availability of clinical data including patient age and follow-up, as well as tumor histological type and grade, clinical stage, and residual tumor size. All tumors were uniformly histopathologically reviewed and re-classified according to new WHO criteria [7,20]. Additionally, a complete evaluation of genetic variants in the TP53 gene (for all tumors) and the TP53 protein status (for cancers only) was performed, as previously described, by either next-generation sequencing (NGS) [21] or with the mouse monoclonal antibody [15]. All the BOTS patients did not undergo any chemical treatment, whereas all OvCa were excised from previously untreated patients. Twenty-two ovarian cancer patients were treated postoperatively with platinum/cyclophosphamide (PC), while eighty-one underwent the taxane/platinum (TP) treatment after a surgical intervention. As to the evaluation of clinical endpoints, all surviving patients had at least a 3-year follow-up. In BOTS, RFS and the presence of microinvasions or non-invasive implants within the tumor masses were used as dependent variables determining the disease prognosis. As the covariates, taken into account in the multivariable statistical inference in BOTS, clinical stage according to FIGO (categorical variable), and patient age (continuous variable) were used. In addition, BOTS were analyzed in the entire cohort of patients, and subgroups comprising specimens with (BOT.V600E) or without (BOT) the *BRAF* V600E mutation, since the presence of this genetic alteration was previously found to be significantly correlated with the lower age of patients diagnosed with BOTS [19]. In cancers, OS and DFS were used as dependent prognostic variables, while PS and CR served as dependent factor variables predictive of response to treatment. CR was defined as the disappearance of all clinical and biochemical symptoms of ovarian cancer assessed after completion of the first-line chemotherapy and confirmed four weeks later. DFS was assessed only for the patients who achieved a CR. As to the independent variables used in multivariable statistical analyses in cancers, a FIGO stage of the tumors along with a residual tumor size were taken into account as factor covariates. Noteworthily, due to the small size of the lgOvCa subgroup, only hgOvCa samples were subjected to regression analyses performed in the present study. In such analyses, hgOvCa were investigated either as the entire group of specimens or in subgroups depending on the chemotherapy regimen used (PC or TP) and/or the TP53 accumulation status. The clinicopathological data were missing for one BOT.V600E specimen. Therefore, the relevant cohort described in Appendix A is smaller.

### 2.2. DNA Isolation and Quality Assessment

Our preliminary analyses revealed that the sample source (snap-frozen or FFPE) significantly affected hierarchical clustering of the data when overall differences in methylation patterns between the specimens were displayed on a heatmap (Appendix A). To eliminate this impact and reduce a potential bias in the methylation analysis results, each group of samples contained DNA isolated from both snap-frozen and FFPE sections (BOT: 4 snap-frozen, 10 FFPE, BOT.V600E: 4 snap-frozen, 7 FFPE, lgOvCa: 5 snap-frozen, 2 FFPE, and hgOvCa: 92 snap-frozen, 4 FFPE). Genomic DNA (gDNA) from snap-frozen sections was isolated using the QIAmp DNA Mini Kit (Qiagen; Hilden, Germany), whereas gDNA from FFPE blocks was extracted in the MagCore Nucleic Acid Extractor, using the MagCore Genomic DNA FFPE One-Step Kit (RBC Biosciences, Xinbei City, Taiwan). Before its hybridization to microarrays, gDNA quality was assessed using our in-house developed method based on the comparison of Real-Time quantitative PCR efficiency for two amplicons of different lengths, described in a paper by Woroniecka et al. [22].

### 2.3. DNA Bisulfite Conversion

High-quality gDNA isolated from tumors was subjected to a bisulfite conversion (EZ DNA Methylation Kit, Zymo Research; Irvine, CA, USA). Before and after the conversion, gDNA concentrations were measured on the Qubit 4 Fluorometer (Thermo Fisher Scientific; Waltham, MA, USA) using either the Qubit dsDNA HS Assay Kit or Qubit ssDNA Assay Kit, respectively (both kits were manufactured by Thermo Fisher Scientific). The bisulfite conversion was carried out for 500–1000 ng of gDNA from snap-frozen tissue sections and 200–1000 ng of gDNA from FFPE blocks.

### 2.4. Microarray Profiling

Bisulfite-converted gDNA samples were subjected to microarray-based DNA methylation profiling with Infinium MethylationEPIC v1.0 BeadChip microarrays (Illumina; San Diego, CA, USA). For identifiers and genomic locations of over 850,000 methylation sites detectable with these microarrays, refer to Appendix A: Illumina_Infinium_methyl_EPIC_array_hg19_ext_attributes.xlsx. Hybridization was carried out according to the protocol provided by Illumina. The fluorescence signal was scanned with the iScan array scanner (Illumina).

### 2.5. Methylation-Specific PCR and Sanger Sequencing

Methylation changes at selected genomic sites were confirmed for three CpGs in three genes by methylation-specific PCR (employing the AmpliTaq Gold™ DNA Polymerase, Thermo Fisher Scientific) followed by Sanger sequencing, using the in-house designed primers: *DHDDS*/*HMGN2* (cg26108329, chr1:g.26797585 and cg05304531, chr1:g.26797576, Forward: TAATATGATTGGGGTATAGTAGAGGTGATT, Reverse: CACTAAATTAATCCCATCTAATTTCTTAAA) and *SKI* (cg13488570, chr1:g.2222253, Forward: TTGTTGAGATATTTTATTGGTTTGAGGGT, Reverse: AACTAATTCACCAAAAATCAAACTCAATTA). Each of the genomic positions mentioned above refers to the GRCh37 assembly of the human genome. PCR products were then analyzed by agarose gel electrophoresis using the Simply Safe reagent (EurX, Gdansk, Poland) for DNA visualization. Gels were documented on the UVP ChemStudio Imaging System (Analytik Jena, Jena, Germany). Afterward, the PCR products were cleaned with ExoSAP-IT (Thermo Fisher Scientific) and subjected to Sanger sequencing using the appropriate primer and the BigDye Terminator v. 3.1 Cycle Sequencing Kit (Thermo Fisher Scientific). Sanger sequencing products were then cleaned with the ExTerminator Kit (A&A Biotechnology, Gdansk, Poland) and analyzed on the 3500 Genetic Analyzer (Thermo Fisher Scientific). The conditions of methylation-specific PCR and Sanger sequencing reactions for each gene are presented in Appendix A.

### 2.6. Bioinformatic and Statistical Analyses

All computations shown herein were run in the R environment (v. 4.3.2), using the GRCh37 (hg19) version of the human genome assembly as a reference. To ensure the highest standards of the methylation analysis, samples with poor hybridization quality were filtered out at the earliest step of the bioinformatic workflow. The hybridization quality was assessed by calculating the signal detection probability with the detectionP function (minfi package, v. 1.46.0). At least 85% of hybridization signals for each sample had to have *p*-values < 0.05 for the sample to remain in the analyses. All our samples passed that filter (Appendix A) and were submitted to the Gene Expression Omnibus (GEO) database (data acc. no. GSE267068). Apart from the samples, hybridization probes also underwent a three-step filtering, involving the detection probability cut-off (*p*-value < 0.05), filters of SNPs at CpG sites, and of cross-reactive probes. Due to the relatively poor quality of DNA isolated from FFPE blocks, we had to eliminate about 24% of the probes at the first filtering step to guarantee reliability of the final results. Therefore, the ultimate number of probes that passed all the filtering steps was 599,503 (69.24%). Subsequent bioinformatic analyses were performed in line with the workflow published by Maksimovic et al. [23] that was further improved by our team, as described in our previous work [19].

All differentially methylated regions (DMRs) identified in our bioinformatic analyses were subsequently subjected to detailed statistical inference with the use of univariable and multivariable Cox proportional hazards models (package: survival, v. 3.5-7) to assess the value of these DMRs as potential novel prognostic biomarkers. All Cox models were also checked with respect to proportionality of hazards for each variable used. The prediction of treatment response was carried out by generating univariable and multivariable logistic regression models (packages: stats. v. 4.0.2, and rms, v. 6.0-1). The dependent, independent, and grouping variables used (different for BOTS and hgOvCa) were described above in the section entitled “Patients and clinicopathological parameters”. In order to verify the discriminative capabilities of the created Cox and logistic regression models, we performed their cross-validation in new data sets, obtained from the original data by bootstrapping (with replacement), using the riskRegression package for R (v. 2023.12.21) [24]. Subsequently, areas under ROC curves (AUCs) between the original and bootstrapped data sets were compared.

To perform detailed gene ontology analyses, each CpG was assigned to the gene only when the CpG site was located on the same DNA strand as the coding sequence of the gene of interest. Furthermore, methylation alterations were analyzed either collectively or with regard to the direction of each change (i.e., hypermethylated genes were assessed independently of hypomethylated ones). The obtained lists of genes were then subjected to ontology analyses with the ShinyGO web app (v. 0.80), with the FDR cutoff set to 0.1 and the maximum pathway size of 2000.

## 3. Results

### 3.1. The Analysis of the MDM2/TP53/CDKN1A (p21) Axis

The *MDM2*/*TP53*/*CDKN1A* axis is a main pathway involved in the determination of genomic stability and the regulation of cell cycle progression [25]. Considering that methylation changes may precede mutations [2], and that the methylome of BOTS and lgOvCa has been poorly investigated so far, we intended to check whether methylation patterns in *TP53* and other genes in the aforementioned axis are different in BOTS and lgOvCa compared to hgOvCa. We focused mainly on methylation changes in promoters and first exons, as such alterations were proven to make the strongest impact on gene expression [26,27]. In Appendix A, the complete list of CpGs in the *TP53*, *MDM2*, and *CDKN1A* gene regions analyzed herein is presented, whereas all significant methylation differences (average beta values) between the analyzed tumors groups for various regions of these genes are shown in Figure 1 and Appendix A.

Overall, in the *TP53* tumor suppressor gene, we observed a tendency towards hypermethylation in carcinomas in comparison with BOTS (Figure 1A–C). Despite the fact that we found no *TP53* missense mutations and TP53 protein accumulation in our low-grade tumors [21], we observed hypermethylation in almost every region of this gene. The methylation of all *TP53* exons and also the first *TP53* exon only was even higher in lgOvCa than in hgOvCa (Figure 1C and Appendix A). For *MDM2*, encoding an oncogenic protein, we observed an opposite effect. In the proximal promoter region of this gene, we found significantly lower methylation in hgOvCa compared to BOTS (Figure 1D). As to *CDKN1A,* which codes for the p21 tumor suppressor protein, we unexpectedly revealed lower methylation levels within the proximal promoter and 1st exon alike in carcinomas compared to BOTS (Figure 1E,F), especially when BOTS without the *BRAF* V600E variant were considered. Interestingly, the first exon of the *CDKN1A* gene was less methylated in lgOvCa than in hgOvCa.

Of note, no methylation differences in either of the above-mentioned three genes between BOT and BOT.V600E tumors were identified in the present study.

### 3.2. Differences in Methylation Patterns Between Groups

The numbers of differentially methylated CpGs and differentially methylated regions (DMRs) in all inter-tumor-group comparisons are shown in Table 1. In general, global genome-wide hypomethylation positively correlated with the increasing aggressiveness of tumors and was especially evident in the hgOvCa group (the highest ratios of downmethylated/upmethylated CpGs and DMRs in hgOvCa vs. all the other tumor groups). Remarkably, the same ratio for the inter-BOTS comparison was also very high, particularly when DMRs were considered. Moreover, BOT.V600E tumors emerged as the group with the lowest number of differentially methylated CpGs and DMRs compared to all the remaining groups. This suggests that extensive hypomethylation of the genome is what distinguishes BOT.V600E from BOT and, when methylome is considered, BOT.V600E tumors might be placed somewhere in-between BOT and OvCa.

### 3.3. CpG Sites with the Most Differentiated Methylation

Based on *p*-values obtained in the differential methylation analysis of individual CpGs, we identified the most differentiated CpG sites for all six inter-tumor-group comparisons. The upset plot demonstrating the numbers of differentially methylated (DM) CpGs in each inter-tumor-group comparison and the numbers of such CpGs for the specific intersection of tumor groups is shown in Figure 2A. In Figure 2B–G, the distribution of M-values for the most DM CpG site in each inter-tumor-group comparison is displayed. Additionally, Figure 2 is supplemented with Table 2, which shows the 10 most significantly differentiating CpGs (and the genes they are located in) for each inter-tumor-group comparison.

DM CpGs distinguishing BOT from BOT.V600E the most occurred in genes involved in cell adhesion (*MIP*, *ODAD3*, *PTPRF* and *ITGA7*), lipid metabolism (*LRP1*, *CBY1*), cell differentiation (*PTPRF*, *CBY1*), apoptosis (*SPRYD4*, *LRP1*), and ER (endoplasmic reticulum)-related processes (*PRKCSH*, *CYB5R4*). One CpG site, cg19623237, was located in an intergenic region.

The CpG differentiating BOT from lgOvCa the most was located in a pseudogene, *NBPF13P*, involved in nervous system development. Some other CpGs/genes differentiating these tumor groups were also engaged in neuronal processes (*ZIC2*, *GNB1L*). However, the biggest group of CpGs with divergent methylation patterns between BOT and lgOvCa lay in genes associated with transcriptional regulation, such as *ZNF585*, *ZNF341*, *ZIC2*, *RECQ25*, *SAP30BP*, and *ETV4*. CpGs in genes participating in mitochondrial processes (*RTL10*, *COX16*, *SYNJ2BP-CO16*) and cell differentiation (*ZIC2*, *ETV4*) were also identified as differentially methylated between BOT and lgOvCa. There was also one CpG, cg10479053, present on the opposite (minus) strand to the coding sequence of the *PSMD3* gene.

In the BOT vs. hgOvCa comparison, the most differentiating CpG, cg18813601, lay in an intergenic region on chromosome 10. Other DM CpGs occurred in genes involved in neuronal processes (*NBPF13P*, *ZIC2*, *SLC4A10*, *DLX6*, and *CSNK1G2*) and cell differentiation/development (*CTBP1*, *DLX6*, *CSNK1G2*). Additionally, some CpGs were found in genes regulating transcription (*ZIC2*, *CTBP1*, *HNRNPA1L2*), Golgi apparatus functioning (*GORASP2*, *CTBP*1), as well as in pseudogenes (*NBPF13P*, *MRPS31P4*).

Interestingly, in the BOT.V600E vs. lgOvCa comparison, we observed some different processes than when BOT were compared to lgOvCa. CpGs with the most divergent patterns were located in genes involved in cell differentiation and development (*FOXA1*, *PLEKHO1*, *TFDP1*, *ZIC2*). In addition, DM CpGs were found in genes related to cell proliferation (*CAMK2N1*, *PVT1*), apoptosis (*FOXA1*, *PLEKHO1*, *PVT1*), adhesion (*PIP5K1C*, *PLEKHO1*, *TTC6*), cell cycle (*FOXA1*, *TFDP1*), lipid metabolism (*CAMK2N1*, *TFDP1*), neuronal processes (*ZIC2*, *CAMK2N1*), and transcription regulation (*FOXA1*, *TFDP1*, *ZIC2*). One CpG site was present in the gene of unknown function (*TMEM104*).

In the BOT.V600E vs. hgOvCa comparison, except for cg06903478 in *AFMID/TK1* and cg00614081 in *CTBP1*, we observed distinct DM CpGs/genes from those differentiating BOT from hgOvCa. Nonetheless, biological processes affected by these epigenetic changes were similar in both comparisons, since cell development/differentiation (genes: *SKI*, *CTBP1*, *TRABD2A*, *CDX2*), transcription (genes: *CDX2*, *CTBP1*), neuronal processes (genes: *TRABD2A*, *SKI*), and Golgi-dependent processes (genes: *CTBP1*, *STX18*) were identified as terms enriched in genes with CpGs most significantly differentiating BOT.V600E from hgOvCa. Two other DM CpGs, cg18813601 and cg19875936, were located in intergenic regions.

The biggest group of DM CpGs between lgOvCa and hgOvCa lay in genes associated with neuronal processes (*KNDC1*, *SEMA6B*, *HTR5A*). Many such CpGs were present on the opposite strand as the coding sequence of known genes (cg15792713, cg11610925, cg14636714, cg07570470, cg19823504, cg05640731, and cg19307500). For detailed information on these and other CpGs described in the present paper, refer to Appendix A Illumina_Infinium_methyl_EPIC_array_hg19_ext_attributes.xlsx.

In order to verify our microarray results and validate the entire bioinformatic workflow, three CpG sites characterized by diverse methylation patterns between the groups of tumors analyzed herein, cg13488570; chr1:g.(+)2222253 in the *SKI*(+) gene and two CpGs in the *DHDDS*(+) gene, cg26108329; chr1:g.(+)26797585 and cg05304531, chr1:g.(+)26797576, were further investigated by methylation-specific PCR and Sanger sequencing. Positive results of this validation are presented in Appendix A.

### 3.4. Ontological Analyses

By using the ShinyGO web app, we performed a detailed ontology analysis for DM CpGs, taking into account not only the DNA strand (+/−) on which each CpG site is located, but also the direction of a methylation change (up- vs. downmethylated CpGs/genes). The results of our gene ontology (GO)-enrichment analysis (categories: biological process (BP), molecular function (MF), cellular compartment (CC)), as well as Molecular Signature Database analysis (MSigDB, Hallmark gene sets) for all inter-tumor-group comparisons, are presented in Appendix A.

In the BP analyses, we observed downmethylation of genes involved in the regulation of cytoskeleton/cell adhesion in BOTS compared to carcinomas (Appendix A). Such processes were also more frequently downmethylated in BOT than in BOT.V600E. By contrast, genes involved in the cell cycle progression and RNA metabolism were upmethylated in BOT compared to BOT.V600E and lgOvCa (Appendix A). Of note, when comparing BOT to hgOvCa, only genes associated with the cell cycle progression were upmethylated in the former group (Appendix A), while the genes involved in RNA metabolism were deregulated in both directions (Appendix A). Altered DNA methylation was also observed in genes encoding proteins regulating the cell cycle when BOT.V600E were compared to lgOvCa, with hypermethylation in the BOT.V600E group (Appendix A). Interestingly, genes linked to RNA processing/metabolism were deregulated in both directions in the BOT.V600E vs. lgOvCa comparison (Appendix A) but only downmethylated in BOT.V600E compared to hgOvCa (Appendix A). In the lgOvCa vs. hgOvCa comparison, only a few cell adhesion-related terms were enriched, and the genes involved in those processes were deregulated in both directions (Appendix A). As for the genes participating in RNA metabolism/processing and the cell cycle regulation, we observed downmethylation in lgOvCa compared to hgOvCa (Appendix A).

As to the genes involved in cell differentiation, development, and morphogenesis, no differences in methylation patterns were found between BOT and BOT.V600E (Appendix A). Simultaneously, genes involved in the aforementioned terms were mainly downmethylated in BOTS compared to carcinomas (Appendix A). Still, in the BOT.V600E vs. hgOvCa comparison, up- and downmethylation were detected at the same time (Appendix A). By contrast, when lgOvCa and hgOVCa were compared to each other, genes associated with differentiation, development, and morphogenesis turned out to be upmethylated in less aggressive tumors (Appendix A). Consistently, the genes responsible for neuronal processes were also upmethylated in lgOvCa compared to aggressive carcinomas (Appendix A). However, when this group of genes was investigated in BOTS, their methylation changes did not differentiate BOT from BOT.V600E (Appendix A). The neuronal processes-related GO terms were, however, deregulated in both ways when BOTS were confronted with hgOvCa (Appendix A). Finally, when compared to lgOvCa, genes related to neuronal processes were downmethylated in BOT and upmethylated in BOT.V600E (Appendix A, respectively).

Methylation alterations in genes associated with intracellular transport were identified when the BOT group was compared to carcinomas with hypomethylation found in more aggressive tumors (Appendix A). Similar regularity was observed in the lgOvCa vs. hgOvCa comparison, where the transport-related terms were enriched in upmethylated genes in lgOvCa (Appendix A). Interestingly, no such GO terms were enriched when BOT.V600E were compared to carcinomas (Appendix A–J). Notably, our results of the GO analysis for the MF and CC categories were consistent with those for BP, presented above (Appendix A).

In the MSigDB analysis, we observed the upmethylation of genes linked to the TP53 pathway, mTORC1 complex, oxidative phosphorylation, and unfolded protein response when BOT (but not BOT.V600E) were compared to other tumor groups (Appendix A). By contrast, the same terms were also enriched in genes downmethylated in lgOvCa compared to hgOvCa (Appendix A). Genes involved in fatty acid metabolism and adipogenesis were hypermethylated in BOT compared to the other groups. Another process worth mentioning, glycolysis, differentiated BOT from BOT.V600E, and the related genes were hypermethylated in the former group (Appendix A). Genes involved in glycolysis were also upmethylated in BOT compared to lgOvCa (Appendix A) and deregulated in both directions when BOT were compared to hgOvCa (Appendix A). As for the molecular signatures distinguishing BOT.V600E from lgOvCa, we observed upmethylation of genes involved in the heme metabolism in the former group (Appendix A). Remarkably, the same term was significantly enriched in the BOT vs. BOT.V600E comparison as well, though the changes in methylation patterns were bidirectional (Appendix A). Another interesting observation refers to angiogenesis, as genes associated with this process were downmethylated in both BOTS groups but only compared to hgOvCa (Appendix A). Lastly, the hypermethylation of genes upregulated by KRAS as well as genes related to epithelial-mesenchymal transition distinguished lgOvCa from hgOvCa only (Appendix A) and did not differentiate BOT from BOT.V600E or BOTS from OvCa (Appendix A–J).

### 3.5. The Most Statistically Significant DMRs

Based on *p*-values, we identified the 10 most significant DMRs for each inter-tumor-group comparison (Table 3). In Figure 3, the best DMR for every comparison is shown, being additionally supplemented with the visualization of DNase I hypersensitive sites (DHSS) as well as transcription factor binding sites (TFBS) to evaluate whether the given DMR is transcriptionally active.

DMRs distinguishing BOT from BOT.V600E the most occurred mainly in genes involved in lipid/steroid/ester metabolism (*NR1H3*, *ACP2*, *ACSS2*, *AKR1D1*) and the cell cycle (*KIF23*, *WEE1*).

Interestingly, all the most significant DMRs discriminating BOT from lgOvCa overlapped the MHC region on chromosome 6 (about 3.5 million bp in length). These DMRs were located in genes linked to the immune response (*HLA-DMA*, *GPANK1*, *LY6G5B*, *TAPBP*, *GNL1*) but also to transcription regulation (*BRD2*, *GTF2H4*, *EHMT2*, *ZBTB22*, *DAXX*, *PHF1*), development and differentiation (*BRD2*, *CSNK2B*, *PPP1R18*, *PHF1*), DNA repair (*MDC1*, *GTF2H4*, *PHF1*), apoptosis (*CSNK2B*, *DAXX*, *NRM*), and neuronal functions (*SLC44A4*, *SYNGAP1*, *CUTA*). Some genes were also associated with cytoskeleton (*TUBB*, *PPP1R18*).

In the BOT vs. hgOvCa comparison, the most significant DMRs occurred mainly in genes participating in transcriptional regulation (*EMX2OS*, *CTBP1*, *PRAME*, *MEIS2*, *ATF6B*, *PITX1*), differentiation and development (*EMX2OS*, *CTBP1*, *MEIS2 PITX1*), and protein folding (*ATF6B*, *FKBPL*, *GORASP2*). A few genes were also involved in the regulation of cytoskeleton (*EHBP1*, *TUBB*) and Golgi apparatus (*CTBP1*, *GORSAP2*), cell cycle (*MDC1*, *FKBPL*), and lipid metabolism (*CPT1B*, *CHKB*).

When comparing BOT.V600E to lgOvCa, we observed some similar processes as for the BOT vs. lgOvCa comparison. Genes linked to immune processes (*ADAP1*, *LTBR*, *TPRG1*) were also identified as differentially methylated, but they were not so abundant. The most significant DMRs were associated with neurological processes (*ADAP1*, *MYRF*, *SCNN1A*), adhesion (*PPP1R18, CAP2B*, *SCNN1A*), and lipid metabolism (*ESR1*, *LTBR*), too.

If BOT.V600E were confronted with hgOvCa, the biggest differences in methylation patterns were found in genes involved in lipid metabolism (*CPT1B*, *CHKB*, *DGKG*), cytoskeletal regulation (*TTLL10*, *NHERF2*), neurological processes (*SKI*, *RPTOR*), differentiation and development (*RPTOR*, *SKI*), and the regulation of transcription (*ZNF551*, *ZNF154*).

Finally, the biggest methylation alterations between lgOvCa and hgOvCa were revealed in genes participating in neuronal processes (*KNDC1*, *EMX2*, *SSTR5*), ubiquitination (*TRIM15*, *TRIM10*, *UNKL*), cytoskeletal regulation/adhesion (*TNXB*, *TRIM15*), differentiation/development (*EMX2*, *TRIM10*), and immune response (*TRIM15*, *TRIM10*).

### 3.6. Cox and Logistic Regression Analyses for DMRs in BOTS and hgOvCa

Each DMR had to be differentially methylated in at least one of six inter-tumor-group comparisons to be subjected to the regression testing, which gave the total number of 128,168 tested DMRs. Uni- and multivariable regression analyses were carried out for all BOTS and hgOvCa available in our sample set. Remarkably, due to the small number of specimens making the multivariable statistical testing impossible, the lgOvCa group was excluded from the regression analysis herein.

To decrease the risk of false-positive hits, we decided to change the statistical significance level (alpha) of our Cox regression models and logistic regression models (lrm) in hgOvCa to 0.0005 and 0.005, respectively. Considering the relatively small size of the BOTS series, the default alpha value of 0.05 was kept in all regression models performed in this series of tumors. To further decrease the risk of obtaining false-positive hits, we focused on those DMRs only for which the results of univariable and multivariable regression tests matched. The models were considered matching when the analyzed DMRs and groups of tumors were the same, both *p*-values < alpha value, both HR/OR values either higher or lower than 1, and concomitantly the discriminative capabilities of all models, uni- and multivariable, before and after a bootstrap-based cross-validation were good enough (all AUC values > 0.7). This approach let us identify 112 and 168 unique matching DMRs in Cox and lrm analyses in hgOvCa, respectively. For BOTS, we obtained 143 matching DMRs, all in the lrm analysis. The detailed results of our regression analyses are provided in Appendix A: Reg.analyses.Cox.hgOvCa.p.val.0.0005.xlsx, Reg.analyses.lrm.hgOvCa.p.val.0.005.xlsx, and Reg.analyses.lrm.BOTS.p.val.0.05.xlsx, collectively abbreviated as Reg.anal.suppl.results. We also performed the GO analysis for all the genes identified in our regression tests as good discriminators in hgOvCa and BOTS. The enriched GO terms along with the genes assigned to each term are provided in Appendix A, respectively. Next, we nominated five DMRs with the lowest *p*-values from each of the three xlsx files as the most promising potential biomarkers in BOTS and hgOvCa. The regression analyses’ results for these DMRs are described below and presented in Table 4, and in Figure 4 and Figure 5. For detailed information on the DMRs listed in this table, including the CpG sites forming each DMR, refer to Appendix A.

In hgOvCa, we managed to identify DMRs predictive of both cancer prognosis and response to chemotherapy. In the former group, all the DMRs were located on chromosome 22, two of them, chr22:g.(−)35776686–35777032 and chr22:g.(−)35775959–35777032, overlapped the *HMOX1* gene, whereas the remaining three, chr22:g.(−)31002067–31003655, chr22:g.both 31002067–31003655, and chr22:g.both 31002362–31004367, encompassed the *TCN2*, *PES1*, and *RP1-56J10.8* genes. Hypermethylation of *HMOX1*-containing DMRs improved the overall survival of hgOvCa patients treated with taxane/platinum (TP), whose tumors exhibited accumulation of the TP53 protein. This favorable factor turned out to be independent of a large residual tumor size, being the marker of poor prognosis. Similarly to the *HMOX1*-overlapping DMRs, those encompassing the *TCN2*, *PES1*, and *RP1-56J10.8* genes, if hypermethylated, were also predictors of good prognosis, and their clinical importance was revealed in the TP-treated patients and/or those with tumors harboring the accumulation of the TP53 protein.

As to the response to chemotherapy, the strongest predictor was a single CpG site, cg10273669, located on the minus strand of chromosome 16, chr16:g.(−)880831–880831. Its hypermethylation increased the chance of tumor complete remission (CR), and this regularity was found in the entire cohort of hgOvCa patients and also in those who underwent the TP treatment. In addition, we identified two other DMRs, *NCAM1(+)/RP11-629G13.1(−)*:chr11:g.(−)112831728–112832249 and *AC006372.4(+)/NA(−)*:chr7:g.(−)157258854–157259343, that could potentially be used to predict the treatment outcome. Hypermethylation in both these regions was recognized herein as the favorable factor increasing the probability of cancer remission. This association was found in the entire cohort of patients and the subgroup treated with TP, as well. When the impact on platinum sensitivity (PS) was considered, two promising potential biomarkers were discovered in our study. The first DMR, *NPTXR(−)/NA(+)*:chr.22:g.(+)39240094–39240424, was located on chromosome 22 and encompassed the *NPTXR* gene. The elevated methylation of CpGs forming this DMR emerged as an advantageous predictive marker, increasing the sensitivity of the tumors to chemotherapy. Its clinical meaning turned out to be independent of the large residual disease, being the factor that significantly worsened cancer prediction. The last DMR, ABR(−)/NA(+):chr17:g.(−)1131424–1131781, being located on chromosome 17 and overlapping the *ABR* gene, was found herein to affect CR and PS alike, and, similarly to other DMRs described in this section, its hyperpermethylation made the hgOvCa tumors more sensitive to chemical treatment in both the entire cohort of patients and also those treated with TP.

In BOTS, no prognostic factors (determining relapse-free surivival (RFS)) were found, but still we managed to identify DMRs potentially suitable as biomarkers predictive of the occurrence of microinvasion and/or non-invasive implants. All these DMRs were discovered in the entire cohort of BOTS patients, irrespective of the presence of the *BRAF* V600E mutation in tumors. The DMR on chromosome 16, *BAIAP3(+)/NA(−)*:chr.16:g.(−)1389301–1389301, containing a single CpG site in the *BAIAP3* gene, cg01881308, may be considered the most promising biomarker in BOTS given the lowest *p*-value of all analyzed DMRs. Remarkably, out of all DMRs presented in this section, this was the only one the hypermethylation of which was a negative predictive factor, elevating the risk that microinvasion or non-invasive implants occur. Methylation changes in all the remaining DMRs in BOTS, listed in Table 4, exhibited a similar clinical effect, as hypermethylation of each of the following regions, *IL34(+)/NA(−)*:chr16:g.both 70613332–70613944; *IL34(+)/NA(−)*:chr16:g.(−)70613332–70613944; *WNT10A(+)/NA(−)*:chr2:g.(+)219748780–219748780; and *NEU1(−)/SLC44A4(−)/NA(+)*:chr.6:g.(+)31827414–31834178, was identified herein as a favorable clinical factor, decreasing the risk of microinvasion and/or non-invasive implants in BOTS. All these four DMRs were found to be potential biomarkers independent of the high FIGO stage, being a strong, negative predictive factor.

## 4. Discussion

In this study, the global genome-wide hypomethylation positively correlated with the increasing aggressiveness of ovarian tumors, being the strongest in hgOvCa. As expected, the *TP53* tumor suppressor gene was hypermethylated in carcinomas compared to BOTS. The methylation was especially high in *TP53* exons in lgOvCa, where no missense mutations were found. Remarkably, all the ten most significant DMRs, discriminating BOT from lgOvCa, encompassed the MHC region on chromosome 6, where genes linked to the immune response are located. Of note, the biggest number of unique DM CpGs and DMRs was found between lgOvCa and hgOvCa, thus corroborating vast methylation differences between these two cancer types reported by others [11]. By contrast, the BOT.V600E tumors had the lowest number of DM CpGs and DMRs compared to all other groups and, in relation to BOT, their genome was strongly downmethylated. This suggests that extensive hypomethylation is what distinguishes BOT.V600E from BOT and, when methylome is considered, BOT.V600E tumors might be placed somewhere in-between BOT and OvCa. By assessing differentially methylated CpGs, we revealed downmethylation of genes involved in the regulation of cytoskeleton/cell adhesion in BOTS compared to carcinomas. Such processes were also more frequently downmethylated in BOT than in BOT.V600E. By contrast, genes involved in cell cycle progression and RNA metabolism were upmethylated in BOT compared to BOT.V600E and lgOvCa. When comparing BOT to hgOvCa, only genes associated with cell cycle progression were upmethylated in the former group. As to the genes involved in cell differentiation, development, and morphogenesis, they were mainly downmethylated in BOTS compared to carcinomas. By contrast, when lgOvCa and hgOvCa were compared, such genes turned out to be upmethylated in less aggressive tumors, suggesting that in highly undifferentiated cancers, likely in the subpopulation of cancer stem cells (CSC), the pathological differentiation to various cell lineages might be advantageous for hgOvCa cells, enabling their epithelial-mesenchymal plasticity [28]. Lastly, in lgOvCa compared to hgOvCa, we detected the hypermethylation of genes upregulated by KRAS as well as genes related to epithelial-mesenchymal transition. These terms did not differentiate either BOT from BOT.V600E or BOTS from OvCa. We also identified hundreds of DMRs in the genome, being of potential use as predictive biomarkers in BOTS and hgOvCa. DMRs with the best discriminative capabilities overlapped the following genes: *BAIAP3*, *IL34*, *WNT10A*, *NEU1*, *SLC44A4*, and *HMOX1*, *TCN2*, *PES1*, *RP1-56J10.8*, *ABR*, *NCAM1*, *RP11-629G13.1*, *AC006372.4*, *NPTXR* in BOTS and hgOvCa, respectively.

Methylation changes are often associated with the initial phase of tumorigenesis and can serve as valuable prognostic and predictive markers [2,29]. Notably, our methylome analyses were performed not only collectively in both DNA strands but also independently in separate strands (either plus or minus) to enhance the precision of the entire workflow. To date, in the literature, there were practically no scientific reports utilizing a similar approach, except for a study demonstrating that CpG methylation solely on the sense DNA strand of the APC gene was specific to hepatocellular carcinoma [30]. Another noteworthy feature of our workflow is its capability to determine methylation alterations in functionally annotated gene regions, including not only coding sequences, but also intron/exon boundaries, introns, UTRs, and proximal and distal promoters. So far, for ovarian tumors, no scientific reports employing such a comprehensive and detailed analytical workflow have been published, which makes our study unique and exceptionally thorough. Alterations in promoter methylation and their influence on gene expression are quite well known in ovarian cancer [27,31]. However, intragenic methylation changes have also been shown to affect transcription. Singer et al. [32] demonstrated two opposite phenomena. Firstly, they observed that some exons are more highly methylated than adjacent introns. Yet, they also identified a subset of mostly hypomethylated exons, which was associated with loose chromatin and thus higher transcriptional activity. Other studies showed that the methylation of first exons [26] and first introns [33] was negatively correlated with transcription, too. Aberrant methylation within the 3′UTRs possibly also affects gene expression, as it was shown that high methylation level of 3′UTRs may stimulate transcription [34]. This outcome, seemingly antithetical with those observed for promoter regions, suggests that the interplay between gene methylation and expression is far more complex and conceivably involves other regulatory processes. In fact, two possible mechanisms might link DNA methylation to gene expression. The first one involves proteins with domains binding to methylated DNA, acting as anchors for other proteins, being gene activity regulators. The second mechanism may rely on changes in DNA properties, such as its affinity to transcription factors and the 3-dimensional structure of chromatin [35]. Furthermore, it needs to be emphasized that gene expression depends not only on DNA methylation alterations but also on other phenomena, e.g., the miRNA-guided transcriptional control [36].

Of note, the results of our methylation analysis within gene regions, for the genes discussed below but not described in the Results section, are available in the Appendix A: GeneRegions.pdf (GR file).

Cancer methylome changes cannot be simply put as either hypo- or hypermethylation of the genome. In fact, both these events occur in malignant cells to some extent, with a tendency towards global, genome-wide hypomethylation in advanced carcinomas. However, hypermethylation of CpG islands associated with, e.g., tumor suppressor genes and developmental regulators is also the hallmark of cancer cells. Additionally, methylation patterns can change dynamically at different stages of tumorigenesis [27,37]. The results shown herein are consistent with those presented in the papers cited above, since we found both hyper- and hypomethylated CpGs and DMRs in our series of cancers, especially in the hgOvCa group, compared to BOTS.

In the results section, we first concentrated on methylation changes within the *MDM2/TP53/CDKN1A* axis, involved in the control of genomic stability [25], as this aspect is still relatively poorly investigated. Despite the fact that we did not find either missense mutations or TP53 protein accumulation in our lgOvCa tumors, we discovered strong hypermethylation of the *TP53* gene in this tumor group compared to all the others. This implies that, in lgOvCa, the activity of the TP53 tumor suppressor may be mainly decreased by epigenetic changes and not mutations. So far, this phenomenon has not been reported by other researchers. Decreased methylation of *MDM2* in hgOvCa probably results in the overexpression of this oncogene, which further impairs the anticancer role of TP53, as MDM2 catalyzes TP53 polyubiquitination, thus causing its degradation in proteasomes [38]. As for another tumor suppressor gene, *CDKN1A*, we expected it to be hypermethylated in OvCa compared to BOTS, as the high *CDKN1A* promoter methylation, leading to its low expression, can help cancer cells evade the cell cycle arrest by diminishing the amount of the p21 tumor suppressor, being a product of this gene [39]. In line with this assumption, the *CDKN1A* promoter hypermethylation was found in various cancers, such as lung, prostate, breast and pancreatic cancer, and leukemia. However, depending on the molecular context, p21 may play either an oncogenic or a tumor-suppressor role [39]. In ovarian cancer cells, especially those harboring the TP53 mutations, the mechanism of *CDKN1A* action may be different, given that the p21 activity depends on TP53 [40]. As shown in our previous study [21], over 60% of hgOvCa samples harbored missense *TP53* mutations, leading mainly to TP53 protein accumulation. By contrast, in lgOvCa, lacking genetic alterations in *TP53*, exceptionally strong hypermethylation of *TP53* was detected herein, as mentioned above. Thus, in both carcinoma groups investigated in this paper, the activity of TP53 seemed substantially impaired which arguably affected its interactions with p21, too. Considering that some genetic alterations in TP53 have previously been reported as gain-of-function, oncogenic mutations [41], it is probable that the role of *CDKN1A* and p21 may also change from anticancer to cancer-promoting when the TP53-dependent molecular context is aberrated. Such a functional shift would explain the negative correlation between *CDNK1A* methylation and tumor aggressiveness revealed in the present study.

Our ontological analyses showed that the processes mainly deregulated in our tumor groups were development/differentiation, adhesion, nervous system, cell cycle, and processes affecting RNA metabolism. Cell differentiation and development are predominantly controlled by transcription co-regulators belonging to the Polycomb group (PcG), including Polycomb Repressive Complex 1 (PRC1) and 2 (PRC2) [42], as well as their targets. One of the PRC targets, the HOXA5 gene, seems to play a significant role in ovarian biology and may be involved in ovarian cancer predisposition, since the loss of *HOXA5* function leads to the formation of ovarian epithelial cysts in older females [43,44]. Moreover, the promoter region of this gene was shown to be hypermethylated in breast cancer [45]. Consistently, in our study, we found high methylation in the coding sequence (cds), distal promoter, and exonic regions of *HOXA5*, especially in hgOvCa but also in BOT.V600E compared to BOT (GR file). In contrast to hypermethylation of Polycomb target genes, in OvCa, we observed hypomethylation of Polycomb genes, as well. In the *EZH2* gene (encoding a protein being a member of the PRC2 complex), especially in hgOvCa compared to BOTS, we detected strong hypomethylation in many regions, including the first exon/5′UTR, exons, and both promoters (GR file). The *BMI1* gene (the protein product of which is a part of the PRC1 complex) was also hypomethylated in many regions not only in hgOvCa (exons, 3′UTR and distal promoter) but in both carcinoma groups (proximal promoter and the first exon), which implies its high expression in OvCa (GR file). Remarkably, one study supports our results, proving that overexpression of *BMI1* in ovarian cancer promotes metastasis, decelerates apoptosis, and desensitizes tumor cells to platinum treatment [46]. Of note, in the present study, the genes involved in cell differentiation, development, and morphogenesis were downmethylated in BOTS compared to carcinomas. By contrast, when lgOvCa and hgOvCa were confronted with each other, such genes turned out to be upmethylated in less aggressive tumors, suggesting that in highly undifferentiated cancers, likely in the subpopulation of CSC, the pathological differentiation to various cell lineages might be advantageous for hgOvCa cells, enabling their epithelial–mesenchymal plasticity [28,47].

Adhesion and cytoskeletal processes were enriched in genes mainly downmethylated in BOTS compared to carcinomas and also when BOT were confronted with BOT.V600E. Two studies employing gene expression microarrays, performed on a small group of cystadenomas, BOTS, and OvCa, all of a serous type, seem to support our results. One of the reports unraveled that the malignant subtype of BOTS exhibited a cell adhesion signature [48], whereas in the other, genes implicated in adhesion, cell cycle, and motility were shown to account for phenotypic differences between borderline tumors and high-grade cancers [49].

GO terms associated with the nervous system were also significantly enriched in differentially methylated genes herein. The literature data are mostly consistent with our results, demonstrating that the pathway enrichment analysis for transcriptomic data revealed neural activities (axon guidance, neurogenesis) as promoters of ovarian cancer progression and indicators of poor prognosis. Moreover, four neural genes (*NTN1*, *UNC5B*, *EFNB2*, and *EFNA5*) were nominated as promising biomarkers and therapeutic targets in ovarian cancer patients [50]. Notably, in our regression analyses for DMRs in hgOvCa, predictive capabilities were not confirmed for any of those genes when methylation changes were considered. Another study unveiled the correlation between the elevated expression of neuronal transcription factor Brn-3a (POU4F1) and the decreased rate of apoptosis in ovarian cancer cells [51]. Consistently, we observed significant hypomethylation in the proximal promoter of *POU4F1* in carcinomas compared to BOTS, as well as in the first exon in hgOvCa in comparison with the remaining groups (GR file). A putative tumor suppressor, *ZIC1*, involved in neurogenesis, dorsal spinal cord development, and maturation of the cerebellum [40] was shown to be hypermethylated and silenced in OvCa. This was correlated with increased proliferation, migration, and invasiveness of tumor cells [52]. Our results align with these findings, as we revealed strong hypermethylation of *ZIC1* in carcinomas compared to BOTS in most regions of the gene (GR file).

As to the cell cycle, hypermethylation of genes coding for cell cycle inhibitors, like p16INK4a (CDKN2A) and p15INK4b (CDKN2B), is a well-known phenomenon*, reported for various tumors [53]*. Conversely, in the present study, we observed hypo- rather than hypermethylation of these two genes in carcinomas compared to BOTS. This seemingly antithetical outcome may be attributed to missense mutations in the *TP53* gene, occurring in hgOvCa. As we demonstrated in our previous research [15,16], the TP53 status can determine the clinical significance of other molecular biomarkers. However, this theory does not explain hypomethylation of those genes in lgOvCa, where neither *TP53* missense variants were found [21] nor TP53 protein accumulation was detected [15]. Nevertheless, as discussed above, the *TP53* methylation was exceptionally high in our lgOvCa series, which suggests that the level of TP53 protein in these tumors was conceivably too low to maintain its tumor suppressor activity.

When comparing BOTS to OvCa, genes involved in RNA transcription, metabolism, and processing were deregulated bidirectionally in our study. As to the transcription-related genes (coding for polymerase II subunits), the *POLR2D* gene was significantly hypomethylated not only in carcinomas (across almost all gene regions, except for 3′UTRs), but also in BOT.V600E (proximal promoter) compared to BOT (GR file). By contrast, some other genes encoding the polymerase II subunits were characterized by higher methylation in carcinomas than in BOTS (e.g., promoters and/or first exons of *POLR2G* and *POLR2L*, as well as the distal promoter of *POLR2C*, and the cds of *POLR2E*, GR file). These findings are supported by the study by Bhandari et al. [54], who revealed overexpression of *POLR2D* in multiple cancers, and also showed the *POLR2L* gene to be hypermethylated in a non-small-cell lung cancer cell line.

RNA metabolism and processing relies mainly on RNA binding proteins (RBPs). The role of genes encoding such proteins, *LUC7L2*, *MRPL46*, *MRPL14*, *PARP4*, *STRAP*, and *PAPOLA*, in ovarian tumor development has already been investigated in the literature [55]. In our study, those genes (except for *PARP4*) were predominantly hypomethylated in carcinomas compared to the BOT groups (GR file). Another RBP-coding gene, *CELF2*, was also downmethylated, in the hgOvCa series tested here and in the majority of OvCa cell lines assessed by Piqué et al. [56]. Nonetheless, in contradiction to these findings, the expression of *CELF2* was shown to positively correlate with better prognosis in ovarian cancer patients [57].

Interestingly, in our ontological analyses, some terms prevailed if a particular tumor group was compared to others, e.g., in the BOT group, fatty acid metabolism and adipogenesis were significantly enriched in hypermethylated genes in all three possible comparisons. This outcome is consistent with the literature, since upregulated lipid metabolic pathways were found to increase lipogenesis and lipolysis via exogenous and endogenous uptakes, thus allowing cancer cells to enhance membrane biogenesis and ATP production, and finally to evade apoptosis. In line with this notion, the researchers showed a high level of lipoproteins in serous hgOvCa and concomitantly increased transfer of cholesterol, phospholipids, and triglycerides to such tumors compared to serous BOTS [58]. Furthermore, the increased rate of fatty acid beta-oxidation leads to higher ATP production and faster cellular lamellipodia formation, which facilitates tumor cell migration and invasion [59].

Notably, two other ontological terms were enriched in our study only if two OvCa groups were compared to each other. One of these terms involved genes upregulated by KRAS, while the other was related to epithelial–mesenchymal transition (EMT). Both these terms were enriched in genes hypermethylated in lgOvCa. Given that there were no KRAS-activating mutations in our hgOvCa [21], hypomethylation of KRAS-dependent genes may be the way for these tumors to induce cell proliferation in the presence of a normal *KRAS* protooncogene. As to the EMT-related genes, their downmethylation in hgOvCa was expected, as it probably increased the aggressiveness, chemoresistance, and potential for metastasis of such cancers, e.g., by the overexpression of Snail transcription factors [47]. Accordingly, when we examined differences of methylation patterns in various regions of the *SNAI2* gene, its distal promoter and the 3′UTR were both hypomethylated in hgOvCa compared to lgOvCa (GR file).

As to the DMRs identified in the present study, all the ten most significant ones, discriminating BOT from lgOvCa, encompassed the MHC region on chromosome 6. The concentration of DMRs within a relatively short fragment of the same chromosome may imply that all these DMRs are located within a single chromatin domain. Such domains were previously shown to be regulated in a coordinated manner in the process of carcinogenesis [60]. The aforementioned region on chromosome 6, comprising approximately 3.5 million bp, is densely packed with immunologically important genes [61]. To date, no studies on this region are available in the literature for BOTS and lgOvCa alike. Still, based on our results, we may assume that the immune system, and possibly also other components of tumor microenvironment, may play a pivotal role in the transition from BOTS to lgOvCa. Yet, to shed more light on this complex process, further in-depth research is necessary.

In BOTS, one of the genes overlapped by DMRs with good discriminative capacities was *BAIAP3*. This TP53-dependent gene encodes a brain-specific angiogenesis inhibitor, involved in the endosome to Golgi retrograde transport [40]. Although there are no data on its role in ovarian tumors, its oncogenic meaning was demonstrated in desmoplastic small-round-cell tumor, an aggressive and rare cancer, in which the ectopic expression of *BAIAP3* dramatically enhanced growth and colony formation in vitro [62]. Our results seem to support that outcome, as we observed hypomethylation of the *BAIAP3* distal promoter and 5′UTR/first exon in hgOvCa compared to BOT and BOT.V600E alike (GR file). Still, if only borderline tumors were considered, high methylation of a one-CpG DMR, located in a *BAIAP3* intron, increased the risk of microinvasion/non-invasive implants in our regression analyses.

By contrast, hypermethylation of two DMRs encompassing the *IL34* gene turned out to be a favorable predictor, herein, decreasing the risk of microinvasion/non-invasive implants, which implies an oncogenic role of IL34 in BOTS. In accordance, the literature portrays IL34 as a cancer-promoting interleukin in OvCa, inducing the formation of tumor-associated macrophages (TAM), being the important part of a tumor microenvironment [63].

As for DMRs of potential use as predictors in hgOvCa, one of them encompassed the *HMOX1* (HO-1) gene. This gene encodes an essential enzyme in heme catabolism [40] and is considered an oncogene, highly expressed in gynecological malignancies, including ovarian, cervical, and endometrial cancers. HO-1 is involved in cell proliferation, metastasis, immune regulation and angiogenesis [64]. Consistently, our regression analyses also contribute to the oncogenic role of *HMOX1*, showing that the elevated methylation level within the discussed DMR was associated with the lower risk of death in patients with tumors harboring the TP53 protein accumulation.

Two other genes, *TCN2*, and *PES1*, were overlapped by three DMRs discovered in the present study. If hypermethylated, all these DMRs emerged as markers of good prognosis in patients suffering from hgOvCa with TP53 accumulation who underwent the TP therapy. Thus, based on our results, *TCN2* and *PES1* might both be regarded as oncogenes. Considering the literature data, the elevated level of *TCN2*, a co-factor taking part in the kobalamin (vitamin B12) transport [40], was associated with the increased risk of thyroid cancer development [65], which supports the outcome obtained in the present study. Similarly, the high expression of the *PES1* gene, encoding a nucleolar protein involved in ribosome biogenesis and DNA replication, was shown to be related to tumor cell proliferation, invasion, and metastasis in multiple types of cancer, including ovarian cancer [66,67], which is concordant with our results, too.

The last gene to be discussed, *ABR*, coding for the protein having the GTPase-activating and the guanine exchange factor (GEF) domains [40], is overlapped by a DMR, hypermethylation of which was demonstrated here as a favorable factor increasing the chance of CR in the entire set of hgOvCa specimens. Thus, the gene in questions appears to be an oncogene in ovarian carcinomas. Our analysis of methylation changes in various functionally annotated gene regions (GR file) constitutes another confirmation of the likely pathogenic role of *ABR* in ovarian tumors, unveiling its hypomethylation in hgOvCa compared to less aggressive tumors in all regions except for 3′UTRs. Conversely, other researchers reported the putative tumor suppressive role of *ABR* in both solid tumors, such as medulloblastoma, astrocytoma, and breast cancer, and in acute myeloid leukemia, too [68]. Remarkably, none of those research were carried out on OvCa, which may explain why their results are inconsistent with ours.

## 5. Limitations of the Study

One of the limitations of our study originates from the fact that we analyzed bulk tumor samples being just a part of the entire tumor microenvironment, the complexity and heterogeneity of which might not have been fully captured due to the constraints of the experimental setup applied herein. Secondly, our research was performed on the retrospective (not prospective) cohort of patients, collected for 20 years, meticulously followed up, and carefully checked for compatibility of all clinicopathological parameters. This approach, though widely used, could introduce some hardly definable biases and limit the ability to control for potential confounding factors. Finally, due to the relatively poor quality of DNA isolated from formalin-fixed, paraffin-embedded (FFPE) blocks, we were forced to discard some hybridization probes (and also the corresponding CpG sites) to guarantee the reliability of the methylome profiling results. Approximately 69% of the probes passed all the filtering steps described in the Methods section. Thus, some potentially important methylation differences may have been missed in the present study.

## 6. Conclusions

Herein, the global genome-wide hypomethylation positively correlated with the increasing aggressiveness of ovarian tumors, being the strongest in hgOvCa. Based on our results, we may also assume that the immune system, and likely other components of tumor microenvironment too, possibly play a pivotal role in the transition from BOTS to lgOvCa. Interestingly, the BOT.V600E tumors had the lowest number of differentially methylated CpGs and DMRs compared to all other groups. Thus, when methylome is considered, such tumors might be placed in-between BOT and OvCa. Moreover, we identified hundreds of DMRs in the genome, being of potential use as predictive biomarkers in BOTS and hgOvCa. Therefore, our research not only forms a groundwork for future studies on ovarian tumor methylome but also, by identifying potential biomarkers, might facilitate the fight against this group of diseases and conceivably improve their outcome.

## Figures and Tables

**Figure 1 cancers-16-03524-f001:**
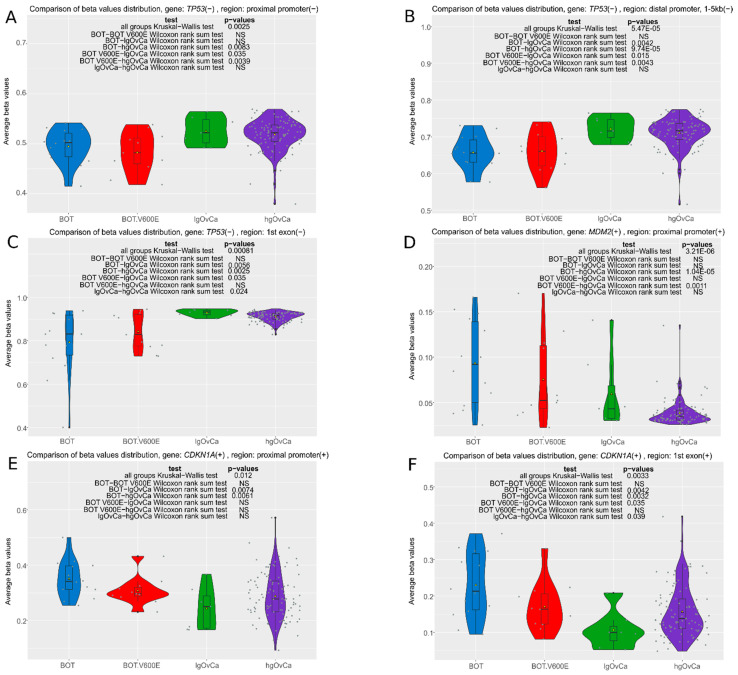
Violin plots of methylation changes (average beta values) in the promoter and first-exon regions of the *TP53*, *MDM2*, and *CDKN1A* genes (the remaining significant results are presented in Appendix A). The values range from 0 to 1 (where 0 means no methylation and 1 denotes 100% methylation of CpGs detected in the region). Each analysis is supplemented with the results of two non-parametric statistical tests: the Kruskal–Wallis test (to determine overall methylation differences between the groups) and the Wilcoxon rank sum test to identify differences between particular groups; NS—non-significant result. Low p-values are displayed in exponential notation (e–n), in which e (exponent) multiplies the preceding number by 10 to the minus nth power.

**Figure 2 cancers-16-03524-f002:**
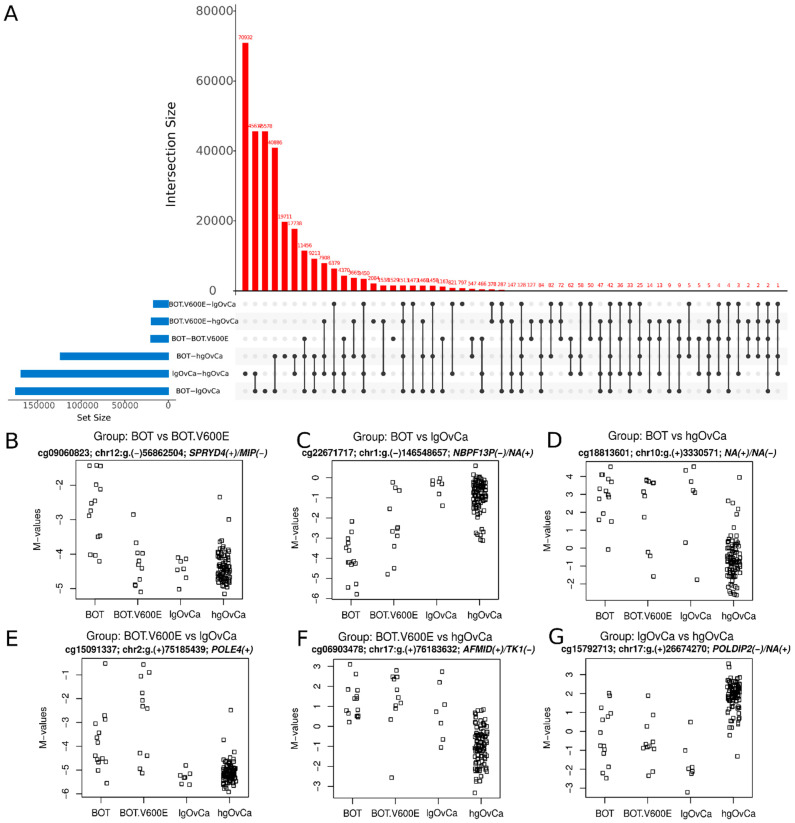
Differentially methylated CpGs. (**A**): the upset plot demonstrating the number of differentially methylated CpGs in each inter-tumor-group comparison (blue bars) and the number of such CpGs (red bars) for the specific intersection of tumor groups (all sets included in the given intersection are indicated with black dots, that are connected with a line if the intersection contains more than one set). (**B**–**G**): the distribution of M-values for the most differentiating CpGs for each inter-tumor-group comparison, followed by genomic locations and gene names with strand identificators shown in brackets. M-value is the log2 of the ratio between signal intensities for probes specific to methylated (numerator) and unmethylated (denominator) cytosines in the given CpG site. The higher the M-value, the higher the methylation level.

**Figure 3 cancers-16-03524-f003:**
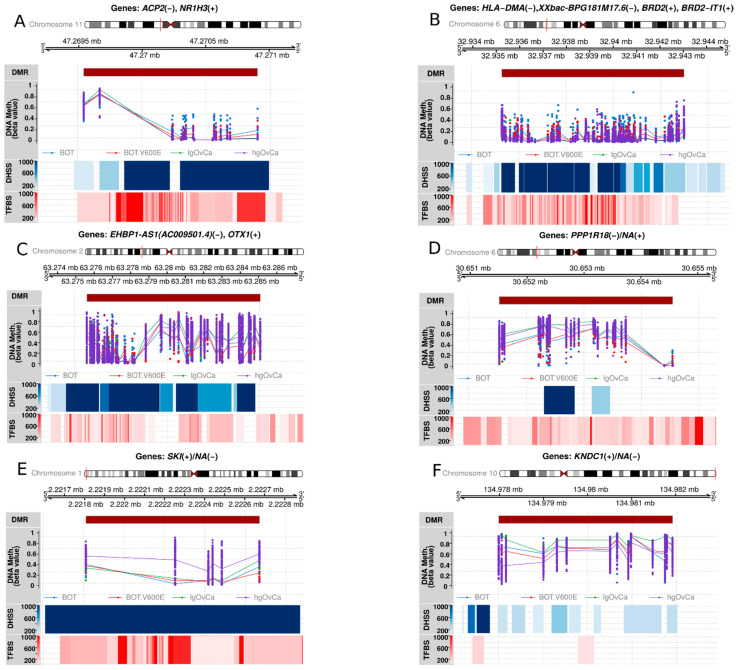
Context plots depicting the most significant DMR for each inter-tumor-group comparison. Each plot title contains encompassed gene name(s) with the DNA strand identifier (+/−), on which the coding sequence of each gene is located. Below, a chromosome ideogram, graphical representation of the genomic range, and DMR location within the genome are shown. These are followed by a line + dot plot demonstrating the distribution of beta values for each CpG and sample (dot) along with mean values for each CpG (line). The visualization of Dnase I hypersensitive sites (DHSS) and transcription factor binding sites (TFBS) is also provided for the assessment of transcriptional activity in each DMR. (**A**): BOT vs. BOT.V600E (chr11:g.both 47269539–47270908); (**B**): BOT vs. lgOvCa (chr6:g.both 32935236–32943025); (**C**): BOT vs. hgOvCa (chr2:g.both 63275602–63285097); (**D**): BOT.V600E vs. lgOvCa (chr6:g.both 30651511–30654559); (**E**): BOT.V600E vs. hgOvCa (chr1:g. 2221807–2222674); (**F**): lgOvCa vs. hgOvCa (chr10:g.both 134977981–134981930).

**Figure 4 cancers-16-03524-f004:**
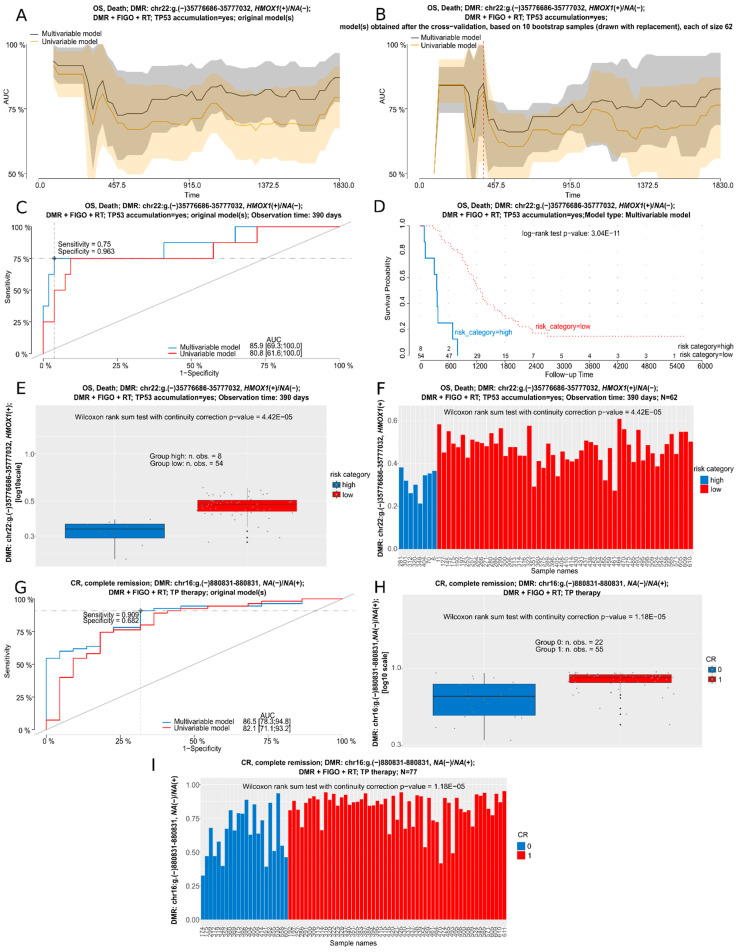
Nominated regression analyses for selected DMRs in hgOvCa. (**A**–**F**): Cox regression analysis (OS) in the subgroup of tumors with TP53 accumulation for the *HMOX1(+)/NA(−)* genes. (**A**,**B**): AUC plot for uni- and multivariable models obtained before (**A**) and after (**B**) a bootstrap-based cross-validation of the original data set. A red dashed line in B indicates the same time point which was used to draw the time-dependent ROC curve (**C**). An optimal cutoff point for this ROC curve, was calculated based on the multivariable model using the Youden index. Discrimination sensitivity and specificity values for this cutoff point are also provided. (**D**): Kaplan-Meier survival curves obtained for the patients divided into two categories (risk higher (high) or lower (low) than for the ROC curve (**C**)-estimated cutoff point) based on the risk of death, calculated using the multivariable model. The Kaplan-Meier curves are supplemented with the result of the log-rank test, as well. Box (**E**) and bar (**F**) plots depicting mean methylation beta values within the DMR in patients with the high or low risk of death. (**G**–**I**): logistic regression analysis (CR) for a DMR in unknown gene(s), in the subgroup of patients treated with the TP regimen. (**G**): ROC curves for uni- and multivariable logistic regression models. Box (**H**) and bar (**I**) plots depicting mean methylation beta values within the DMR in patients with (1) and without (0) CR. RT: residual tumor; TP: taxane/platinum chemotherapy; CR: complete remission. Low p-values are displayed in exponential notation (e−n), in which e (exponent) multiplies the preceding number by 10 to the minus nth power.

**Figure 5 cancers-16-03524-f005:**
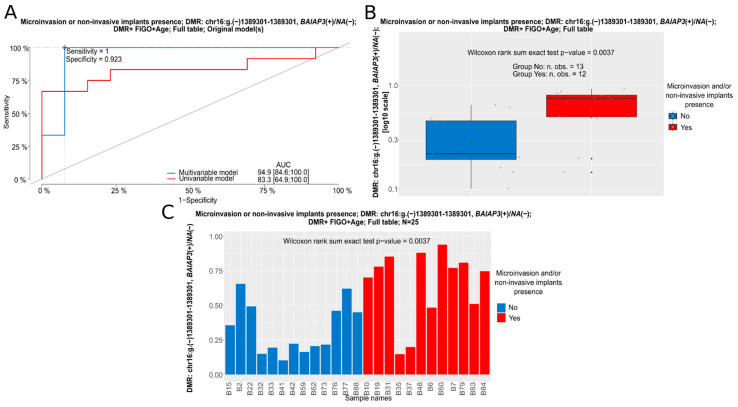
A nominated logistic regression analysis for a DMR in the *BAIAP3(+)/NA(−)* gene in the whole group of BOTS patients (Full table). (**A**): ROC curves for uni- and multivariable logistic regression models; Box (**B**) and bar (**C**) plots depicting mean methylation beta values within the DMR in tumors with (Yes) and without (No) microinvasion/non-invasive implants.

**Table 1 cancers-16-03524-t001:** Numbers of differentially methylated CpGs and DMRs between the groups of tumors.

**CpGs**						
**DM CpGs**	**BOT vs. BOT V600E**	**BOT vs. lgOvCa**	**BOT vs. hgOvCa**	**BOT V600E vs. lgOvCa**	**BOT V600E vs. hgOvCa**	**lgOvCa vs. hgOvCa**
**Upmethylated**	16,108	86,834	93,667	5438	12,170	136,293
**Downmethylated**	4035	88,467	30,227	11,665	7369	32,832
**Sum of DM CpGs**	20,143	175,301	123,894	17,103	19,539	169,125
**NS**	579,360	424,202	475,609	582,400	579,964	430,378
**Up/Down ratio**	3.99	0.98	3.1	0.47	1.65	4.15
**DMRs**						
**DMRs**	**BOT vs. BOT V600E**	**BOT vs. lgOvCa**	**BOT vs. hgOvCa**	**BOT V600E vs. lgOvCa**	**BOT V600E vs. hgOvCa**	**lgOvCa vs. hgOvCa**
**Upmethylated**	1837	12,438	11,442	1062	2127	21,555
**Downmethylated**	25	7646	1979	869	1385	5759
**Sum of DMRs**	1862	20,084	13,421	1931	3512	27,314
**Up/Down ratio**	73.48	1.63	5.78	1.22	1.54	3.74

The up/down prefixes refer to the first element in each comparison. DM—differentially methylated; DMR—differentially methylated region; NS—non-significant.

**Table 2 cancers-16-03524-t002:** CpG sites with the most differentiated methylation in all inter-tumor-group comparisons.

**BOT vs. BOT.V600E**	**BOT vs. lgOvCa**	**BOT vs. hgOvCa**
cg09060823; chr12:g.(−)56862504 ***SPRYD4(+)/MIP(−)***	cg22671717; chr1:g.(−)146548657 ***NBPF13P(−)/NA(+)***	cg18813601; chr10:g.(+)3330571 ***NA(+)/NA(−)***
cg00598858; chr19:g.(−)11545966 ***PRKCSH(+)/ODAD3 (CCDC151)(−)***	cg06869971; chr15:g.(−)69706519 ***KIF23(+)/RP11-253M7.1 (KIF23-AS1)(−)***	cg25977528; chr13:g.(+)100633444 ***ZIC2(+)***
cg24443198; chr6:g.(−)84569302 ***CYB5R4(+)/NA(−)***	cg22011361; chr14:g.(−)70821355 ***COX16(−)/SYNJ2BP-COX16(−)***	cg00614081; chr4:g.(−)1233439 ***CTBP1(−)***
cg10664618; chr12:g.(+)57579466 ***LRP1(+)***	cg25977528; chr13:g.(+)100633444 ***ZIC2(+)***	cg06903478; chr17:g.(+)76183632 ***AFMID(+)/TK1(−)***
cg15086746; chr1:g.(−)44084965 ***PTPRF(+)/NA(−)***	cg03751813; chr19:g.(−)37701393 ***ZNF585B(−)***	cg02608914; chr2:g.(−)171784720 ***GORASP2(+)/NA(−)***
cg00500457; chr22:g.(−)39055589 ***CBY1(+)/FAM227A(−)***	cg23639257; chr17:g.(−)73663270 ***RECQL5(−)/SAP30BP(+)***	cg22671717; chr1:g.(−)146548657 ***NBPF13P(−)/NA(+)***
cg08427970; chr10:g.(−)99122398 ***RRP12(−)***	cg10479053; chr17:g.(−)38136919 ***PSMD3(+)/NA(−)***	cg11704490; chr2:g.(−)162284894 ***NA(−)/SLC4A10(+)/AC009487.5(+)***
cg02608656; chr12:g.(+)56090830 ***ITGA7(−)/NA(+)***	cg17908846; chr20:g.(+)32320553 ***ZNF341(+)***	cg10659805; chr7:g.(+)96631680 ***DLX6(+)/DLX6-AS1(−)***
cg02901790; chr8:g.(+)144391601 ***TOP1MT(−)/NA(+)***	cg22437020; chr17:g.(−)41623744 ***ETV4(−)/RP11-392O1.4(+)***	cg02215357; chr13:g.(−)53191046 ***NA(−)/HNRNPA1L2(+)/MRPS31P4(+)***
cg19623237; chr17:g.(+)77818582 ***NA(+)/NA(−)***	cg00528793; chr22:g.(−)19842837 ***GNB1L(−)/RTL10 (C22Orf29)(−)***	cg25899337; chr19:g.(−)1970441 ***CSNK1G2(+)/NA(−)***
**BOT.V600E vs. lgOvCa**	**BOT.V600E vs. hgOvCa**	**lgOvCa vs. hgOvCa**
cg15091337; chr2:g.(+)75185439 ***POLE4(+)***	cg06903478; chr17:g.(+)76183632 ***AFMID(+)/TK1(−)***	cg15792713; chr17:g.(+)26674270 ***POLDIP2(−)/NA(+)***
cg13518540; chr17:g.(+)72781248 ***TMEM104(+)***	cg27641801; chr4:g.(−)4429265 ***STX18(−)***	cg11610925; chr10:g.(−)134978049 ***KNDC1(+)/NA(−)***
cg00376288; chr19:g.(+)3656580 ***PIP5K1C(−)/NA(+)***	cg08271229; chr1:g.(+)2222674 ***SKI(+)***	cg00454305; chr16:g.(−)1429905 ***UNKL(−)***
cg10168722; chr14:g.(−)38068608 ***FOXA1(−)/TTC6(+)***	cg18813601; chr10:g.(+)3330571 ***NA(+)/NA(−)***	cg18468569; chr8:g.(+)125984720 ***ZNF572(+)***
cg11199810; chr1:g.(−)150123146 ***PLEKHO1(+)/NA(−)***	cg17026391; chr11:g.(+)61159442 ***TMEM216(+)***	cg14636714; chr10:g.(−)135018298 ***KNDC1(+)/NA(−)***
cg18656829; chr13:g.(−)100632250 ***NA(−)/ZIC2(+)***	cg00614081; chr4:g.(−)1233439 ***CTBP1(−)***	cg07570470; chr8:g.(+)142318841 ***NA(+)/SLC45A4(−)***
cg02941008; chr1:g.(+)20810527 ***CAMK2N1(−)/NA(+)***	cg00817355; chr2:g.(−)85073409 ***TRABD2A(−)***	cg19823504; chr19:g.(+)4556982 ***SEMA6B(−)/NA(+)***
cg27641801; chr4:g.(−)4429265 ***STX18(−)***	cg15792713; chr17:g.(+)26674270 ***POLDIP2(−)/NA(+)***	cg21633143; chr7:g.(−)154862021 ***HTR5A(+)/HTR5A-AS1(−)***
cg07819108; chr8:g.(+)128921817 ***PVT1(+)***	cg05222982; chr13:g.(+)28545214 ***NA(+)/CDX2(−)***	cg05640731; chr10:g.(−)135018226 ***KNDC1(+)/NA(−)***
cg17707487; chr13(+)114261869 ***TFDP1(+)***	cg19875936; chr12:g.(−)7858848 ***NA(−)/NA(+)***	cg19307500; chr19:g.(−)1083193 ***HMHA1 (ARHGAP45)(+)/NA(−)***

Names of genes in which the given CpG sites are located including the coding DNA strand (+/−) are emboldened. Overlapping genes are separated with a slash (/). CpG sites’ identifiers and their chromosomal locations, including the strand they lie on, are shown above the gene name and are not emboldened.

**Table 3 cancers-16-03524-t003:** The most significant differentially methylated regions (DMRs) in all inter-tumor-group comparisons.

**BOT vs. BOT.V600E**	**BOT vs. lgOvCa**	**BOT vs. hgOvCa**
chr11:g.both 47269539–47270908; ***NR1H3(+)/ACP2(−)***	chr6:g.both 32935236–32943025; ***BRD2(+)/BRD2-IT1(+)/XXbac-BPG181M17.6(−)/HLA-DMA(−)***	chr2:g.both 63275602–63285097; ***EHBP1-AS1(AC009501.4)(−)/OTX1(+)***
chr6:g.both 31762409–31763873; ***VARS1(−)/NA(+)***	chr6:g.both 30684340–30690844; ***TUBB(+)/MDC1(−)***	chr6:g.both 30683787–30690844; ***TUBB(+)/MDC1(−)***
chr15:g.(−)69706375–69707291; ***KIF23(+)/RP11-253M7.1(KIF23-AS1)(−)***	chr6:g.both 31626915–31634890; ***C6orf47(−)/C6orf47-AS1(+)/CSNK2B(+)/GPANK1(−)/LY6G5B(+)***	chr10:g.both 119291766–119296942; ***EMX2OS(−)/NA(+)***
chr6:g.(−)31762409–31763873; ***VARS1(−)/NA(+)***	chr6:g.both 30874989–30886161; ***GTF2H4(+)/VARS2(+)/NA(−)***	chr4:g.both 1232112–1236678; ***CTBP1(−)/NA(+)***
chr20:g.both 33459881–33461321; ***ACSS2(+)/GGT7(−)***	chr6:g.(+)32935236–32943025; ***BRD2(+)/BRD2-IT1(+)/XXbac-BPG181M17.6(−)/HLA-DMA(−)***	chr22:g.both 22899991–22902665; ***IGL locus (+): LL22NC03-63E9.3(+)/PRAME(−)***
chr12:g.both 7282081–7283890; ***CLSTN3(+)/RBP5(−)/RP11-273B20.1(−)***	chr6:g.both 31850189–31857100; ***SLC44A4(−)/EHMT2(−)/EHMT2-AS1(+)***	chr15:g.both 37391121–37395115; ***MEIS2(−)/RP11-128A17.1(+)***
chr7:g.both 137686266–137687260; ***AKR1D1(+)/CREB3L2(−)***	chr6:g.both 33279563–33287809; ***TAPBP(−)/ZBTB22(−)/DAXX(−)/ NA(+)***	chr6:g.both 32094845–32098253; ***ATF6B(−)/FKBPL(−)/NA(+)***
chr6:g.both 32861863–32862953; ***LOC100294145(+)/HLA-Z(+)NA(−)***	chr6:g.both 30519312–30525976; ***GNL1(−)/PRR3(+)***	chr22:g.both 51016386–51017723; ***CPT1B(−)/CHKB-CPT1B(−)/CHKB-DT(+)/CHKB(−)***
chr11:g.both 9595191–9596475; ***WEE1(+)/NA(−)***	chr6:g.both 30651511–30659692; ***PPP1R18(−)/NRM(−)/NA(+)***	chr2:g.both 171784610–171786316; ***GORASP2(+)/NA(−)***
chr11:g.(+)47269539–47270669; ***NR1H3(+)/ACP2(−)***	chr6:g.both 33381680–33387205; ***PHF1(+)/SYNGAP1(+)/CUTA(−)***	chr5:g.both 134362967–134369605; ***PITX1(−)/PITX1-AS1(+)***
**BOT.V600E vs. lgOvCa**	**BOT.V600E vs. hgOvCa**	**lgOvCa vs. hgOvCa**
chr6:g.both 30651511–30654559; ***PPP1R18(−)/NA(+)***	chr1:g.both 2221807–2222674; ***SKI(+)/NA(−)***	chr10:g.both 134977981–134981930; ***KNDC1(+)/NA(−)***
chr6:g.both 31733434–31734580; ***VWA7(−)/SAPCD1-AS1(−)/NA(+)***	chr19:g.both 58220080–58220818; ***ZNF551(+)/AC003006.7(+)/ZNF154(−)***	chr6:g.both 32044869–32057846; ***TNXB(−)/RNA5SP206(−)/NA(+)***
chr1:g.both 19664276–19665757; ***CAPZB(−)/NA(+)***	chr1:g.both 1102276–1106175; ***MIR200B(+)/MIR200A(+)/MIR429(+)/TTLL10(+)/RP11-465B22.8(+)/NA(−)***	chr6:g.both 30127760–30132715; ***TRIM15(+)/TRIM10(−)***
chr6:g.both 152127812–152129791; ***ESR1(+)/ NA(−)***	chr17:g.both 78865087–78866579; ***RPTOR(+)/NA(−)***	chr19:g.both 405795–409510; ***C2CD4C(−)/NA(+)***
chr7:g.both 964629–967277; ***ADAP1(−)/NA(+)***	chr22:g.both 51016386–51017723; ***CPT1B(−)/CHKB-CPT1B(−)/CHKB-DT(+)/CHKB(−)***	chr10:g.both 119291766–119297716; ***EMX2OS(−)/EMX2(+)***
chr11:g.61521905–61523045; ***MYRF(+)/MYRF-AS1(−)/RP11-467L20.10(−)***	chr16:g.2082689–2083393; ***NHERF2(SLC9A3R2)(+)/NA(−)***	chr12:g.both 132686912–132689907; ***GALNT9(−)/NA(+)***
chr3:g.both 129692836–129694665; ***TRH(+)/NA(−)***	chr19:g.(+)58220080–58220818; ***ZNF551(+)/AC003006.7(+)/ZNF154(−)***	chr12:g.both 132847907–132856142; ***LOC100130238(+)/GALNT9(−)/RP13-895J2.3(+)***
chr12:g.both 6483708–6487080; ***LTBR(+)/SCNN1A(−)***	chr3:g.both 185911208–185912486; ***DGKG(−)/NA(+)***	chr4:g.both 100571622–100574653; ***NA(+)/C4orf54(−)***
chr3:g.both 188664632–188666540; ***TPRG1(+)/TPRG1-AS1(−)***	chr16:g.(−)2082745–2083178; ***NHERF2(SLC9A3R2)(+)/NA(−)***	chr16:g.both 1127792–1132709; ***SSTR5(+)/SSTR5-AS1(−)***
chr3:g.(+)129692836–129694665; ***TRH(+)/NA(−)***	chr1:g.(+)1102276–1106175; ***MIR200B(+)/MIR200A(+)/MIR429(+)/TTLL10(+)/RP11-465B22.8(+)/NA(−)***	chr16:g.both 1428639–1430367; ***UNKL(−)/NA(+)***

Names of genes encompassed by the given DMR, including the DNA strand (+/−) on which the coding sequence of the gene is located, are emboldened. Overlapping genes are separated with a slash (/). A chromosomal localization for each DMR, along with the information whether the DMR was calculated for the plus (+), minus (−) or both DNA strands, is shown above gene name(s).

**Table 4 cancers-16-03524-t004:** The selected results of multivariable Cox and logistic regression analyses for DMRs with the best discriminative capabilities in hgOvCa and BOTS.

**hgOvCa**						
**Cox Regression (alpha = 0.0005)**			**Mean beta Value (%) for DMR**			
**OS in the TP53 Accumulation = Yes Subgroup**	**HR [95% Cl]**	***p*-Value**	**BOT**	**BOT V600E**	**lgOvCa**	**hgOvCa**
***HMOX1(+)/NA(−)*:** **chr22:g.(*−*)35776686–35777032 ^a^**	8.4 × 10^−5^ [0–0.005]	4.11 × 10^−6^	51.05	54.64	49.59	45.18
Residual tumor > 2 cm vs. 0 cm	6.24 [2.315–16.823]	0.0003				
**OS in the TP therapy and TP53 accumulation = yes subgroup**						
***HMOX1(+)/NA(−)*:** **chr22:g.(*−*)35775959–35777032 ^b^**	3.71 × 10^−6^ [0–0.001]	4.33 × 10^−6^	63.24	66.81	65.14	60.05
Residual tumor > 2 cm vs. 0 cm	8.3 [2.525–27.269]	0.0005				
***TCN2(+)/PES1(−)/RP1-56J10.8(+)*:** **chr22:g.(−)31002067–31003655 ^c^**	1.13 × 10^−7^ [0–0]	5.26 × 10^−6^	36.66	32.48	31.57	26.55
***TCN2(+)/PES1(−)/RP1-56J10.8(+)*:** **chr22:g.both 31002067–31003655 ^c^**	4.06 × 10^−11^ [0–0]	6.35 × 10^−6^	27.65	23.92	22.3	18.88
***TCN2(+)/PES1(−)/RP1-56J10.8(+)*:** **chr22:g.both 31002362–31004367 ^c^**	1.15 × 10^−9^ [0–0]	7.31 × 10^−6^	31.29	27.29	25.84	22.48
**Logistic regression (alpha = 0.005)**			**Mean beta value (%) for DMR**			
**CR in the TP therapy subgroup**	**OR [95% Cl]**	***p*-value**	**BOT**	**BOT V600E**	**lgOvCa**	**hgOvCa**
***NA(−)/NA(+)*:** **chr16:g.(−)880831–880831**	5.14 [2.207–11.957]	0.00015	83.21	85.29	92.53	77.47
**CR in the whole group (full table)**						
***ABR(−)/NA(+)*:** **chr17:g.(−)1131424–1131781 ^d^**	7.86 [2.566–24.063]	0.00031	31.71	26.14	38	21.59
***NA(−)/NA(+)*:** **chr16:g.(−)880831–880831**	3.4 [1.72–6.707]	0.00043	83.21	85.29	92.53	77.47
***NCAM1(+)/RP11-629G13.1(−)*:** **chr11:g.(−)112831728–112832249 ^c^**	4.77 [1.975–11.535]	0.00052	28.82	19.77	17.35	13.42
***AC006372.4 (+)/NA(−)*:** **chr7:g.(−)157258854–157259343 ^c^**	5.54 [2.104–14.596]	0.00053	57.89	60.4	65.37	42.17
**PS in the whole group (full table)**						
***NPTXR(−)/NA(+)*:** **chr.22:g.(+)39240094–39240424**	4.04 [1.81–9.03]	0.00066	13.97	8.24	3.42	3.86
Residual tumor > 2 cm vs. 0 cm	0.042 [0.006–0.294]	0.0014				
**BOTS**						
**Logistic regression (alpha = 0.05)**			**Mean beta value (%) for DMR**			
**The presence of microinvasion and/or non-invasive implants in the whole group (full table)**	**OR [95% Cl]**	***p*-value**	**BOT**	**BOT V600E**	**lgOvCa**	**hgOvCa**
***BAIAP3(+)/NA(−)*:** **chr.16:g.(−)1389301–1389301**	49.04 [1.863–1290.778]	0.02	45.4	52.22	63.23	38.27
***IL34(+)/NA(−)*:** **chr16:g.both 70613332–70613944**	0.168 [0.037–0.769]	0.022	52.68	49.89	42.6	47.9
**FIGO II/III vs FIGO IA/IB**	185.5 [2.166–15883.94]	0.021				
***IL34(+)/NA(−)*:** **chr16:g.(−)70613332–70613944**	0.139 [0.025–0.759]	0.023	54.96	51.69	47.07	50.71
FIGO II/III vs. FIGO IA/IB	117.39 [1.936–7116.43]	0.023				
***WNT10A(+)/NA(−)*:** **chr2:g.(+)219748780–219748780**	0.14 [0.025–0.762]	0.023	41.98	36.23	44.2	30.48
FIGO II/III vs. FIGO IA/IB	157.11 [1.691–14593.4]	0.029				
***NEU1(−)/SLC44A4(−)/NA(+)*:** **chr.6:g.(+)31827414–31834178**	0.022 [0.001–0.601]	0.024	53.63	53.38	53.2	53.14
FIGO II/III vs. FIGO IA/IB	569.6 [1.093–296737.5]	0.047				

OS—overall survival; HR—hazard ratio; OR—odds ratio; CR—complete remission; PS—platinum sensitivity; TP—taxane/platinum chemotherapy; ^a^—the same regularity was found in the subgroup: TP therapy and TP53 accumulation = yes; ^b^—the same regularity was found in the subgroup: TP53 accumulation = yes; ^c^—the same regularity was found in the TP-treated subgroup; ^d^—the same regularity was found for CR in the TP-treated subgroup and for PS in both the whole group and the TP-treated subgroup. The missing models can be found in Reg.anal.suppl.results.

## Data Availability

All data are available in the main text or the Appendix A.

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
