# Peer review of "The Diversity of Methylation Patterns in Serous Borderline Ovarian Tumors and Serous Ovarian Carcinomas"

_cancers, 2024, doi:10.3390/cancers16203524_

Round 1
Reviewer 1 Report
Comments and Suggestions for Authors
In this manuscript by Szafron et al entitled “The diversity of methylation patterns in serous borderline ovarian tumours and serous ovarian carcinomas”, the authors aim to explore changes in DNA methylation in ovarian tumours, comparing high-grade, low-grade and borderline ovarian tumours by methylation microarrays. They found variation in the numbers of differentially methylated regions as well as potential correlation with the increasing aggressiveness of the tumours. Their work further shows that specific chromosomal locations as well as specific genes were affected, suggested these targets could be considered as potential biomarkers for the progression of the disease.
The work presented is of interest for the broad cancer community and suggest some possible new avenues of work to be characterised further. There are however a few points, listed below, which should be considered to strengthen the delivery of the work presented.
• The authors provide a large amount of supplementary files either tables (tables S1-S7) and figures (Figures S1-S7) but only a small subset of them are referenced in the text (supplementary Figure S1 and supplementary Table S4). It is important that the authors integrate the remaining supplementary information into the manuscript so that readers can be directed to the data or the supplementary material should be removed if not needed.
• Much work is provided to demonstrate changes in methylation but in the end, there are no direct evidence linking such modification to actual expression changes of the targeted regions/genes. It is obviously a very hard task to request actual data which clearly shows that expression levels have been affected but perhaps the authors could think of ways to support their data to relate this to biological changes.
Author Response
Dear Reviewer,
We greatly appreciate Your interest in this manuscript and would like to thank You for the suggestions and remarks that indeed increased the quality of the paper.
Ad 1. The authors provide a large amount of supplementary files either tables (tables S1-S7) and figures (Figures S1-S7) but only a small subset of them are referenced in the text (supplementary Figure S1 and supplementary Table S4). It is important that the authors integrate the remaining supplementary information into the manuscript so that readers can be directed to the data or the supplementary material should be removed if not needed.
Actually, all supplementary figures were referenced in the manuscript. However, the Reviewer might have been misled by the fact that our paper was initially prepared with the Materials and Methods section placed at the end of the manuscript, and the order of sections was later changed by the editorial office during the initial formatting of the text, so that it meet the requirements of the Cancers journal. As a result, the Materials and Methods section was moved directly after the Introduction which made the order of supplementary figures and tables inconsistent with the main text. Now, after amendments, all data referenced in the text, either in the manuscript or in the supplement, are shown in the order of their appearance in the text. This makes it easier for the reader to follow the main plot and quickly find all necessary information.
Ad 2. Much work is provided to demonstrate changes in methylation but in the end, there are no direct evidence linking such modification to actual expression changes of the targeted regions/genes. It is obviously a very hard task to request actual data which clearly shows that expression levels have been affected but perhaps the authors could think of ways to support their data to relate this to biological changes.
This is a very apt remark. In fact, we are currently studying ovarian tumor transcriptomes using next-generation sequencing. It is worth noting that in the mentioned research we not only assess the levels of expression for particular genes, but we also try to uncover the differences in gene expression patterns between the tumors and the adjacent normal tissue. To perform this analysis as precisely as possible, we apply the Visium spatial transcriptomics technology offered by 10x Genomics, and to push this technique to its limits and obtain both gene expression and sequence variant data at the same time, we combine the Visium workflow with nanopore sequencing on sequencers manufactured by Oxford Nanopore Technologies. As soon as we get the data, we will try to merge the methylome and transcriptome results to have an even deeper insight into the molecular mechanisms of ovarian carcinogenesis.
Yours sincerely,
Lukasz M. Szafron, PhD
Reviewer 2 Report
Comments and Suggestions for Authors
You did a complete analysis of the methylation status on different cancerous tissues. The introduction is enough to understand, and follow, the study for the readers. The methods are well-designed and explained. The results section is difficult to follow due to the amount of information presented, but still, it can be understood. Could it be improved?
In both, the results and discussion sections, there is a bias in the description and discussion of the results proposing that the main (or only) effect of the methylation is on the adjacent regions. This is true, probably, for the promoter regions, but it is less clear for other regions. What about the known influence on the regulation of gene expression in more distant regions due to 3D arrangements of the genome? In this context, how relevant is the localization of the methylation changes on the MHC?
It would help if you made clear that the changes in gene expression are not caused simply by changes in methylation, there are other well-known regulators already demonstrated in cancer tissues.
Author Response
Dear Reviewer,
We greatly appreciate Your interest in this manuscript and would like to thank You for the suggestions and remarks that indeed increased the quality of the paper.
Ad 1. The introduction is enough to understand, and follow, the study for the readers. The methods are well-designed and explained. The results section is difficult to follow due to the amount of information presented, but still, it can be understood. Could it be improved?
It seems possible that the Reviewer might have found it a bit difficult to navigate within the results section due the fact that our paper was initially prepared with the Materials and Methods section placed at the end of the manuscript, and the order of sections was later changed by the editorial office during the initial formatting of the text, so that it meet the requirements of the Cancers journal. As a result, the Materials and Methods section was moved directly after the Introduction which made the order of supplementary figures and tables inconsistent with the main text. Now, after amendments, all data referenced in the paper, either in the manuscript or in the supplement, are shown in the order of their appearance in the text. This makes it easier for the reader to follow the main plot and quickly find all necessary information.
Ad 2. In both, the results and discussion sections, there is a bias in the description and discussion of the results proposing that the main (or only) effect of the methylation is on the adjacent regions. This is true, probably, for the promoter regions, but it is less clear for other regions. What about the known influence on the regulation of gene expression in more distant regions due to 3D arrangements of the genome? In this context, how relevant is the localization of the methylation changes on the MHC?
It would help if you made clear that the changes in gene expression are not caused simply by changes in methylation, there are other well-known regulators already demonstrated in cancer tissues.
The text of our manuscript was modified to address these two remarks of the Reviewer. After amendments, it reads (lines: 654-665):
“Aberrant methylation within the 3’UTRs possibly also affects gene expression, as it was shown that high methylation level of 3’UTRs may stimulate transcription [33]. This outcome, seemingly antithetical with those observed for promoter regions, suggests that the interplay between gene methylation and expression is far more complex and conceivably involves other regulatory processes. In fact, two possible mechanisms might link DNA methylation to gene expression. The first one involves proteins with domains binding to methylated DNA, acting as anchors for other proteins, being gene activity regulators. The second mechanism may rely on changes in DNA properties, such as its affinity to transcription factors and the 3-dimensional structure of chromatin [34]. Furthermore, it needs to be emphasized that gene expression depends not only on DNA methylation alterations but also on other phenomenons, e.g., the miRNA-guided transcriptional control [35].”.
And also (lines 820-825 in the revised version of the manuscript):
“As to the DMRs identified in the present study, all ten the most significant ones, discriminating BOT from lgOvCa, encompassed the MHC region on chromosome 6. The concentration of DMRs within a relatively short fragment of the same chromosome may imply that all these DMRs are located within a single chromatin domain. Such domains were previously shown to be regulated in a coordinated manner in the process of carcinogenesis [59].”.
Yours sincerely,
Lukasz M. Szafron, PhD